# LLMs Can Evolve Continually on Modality for $\mathbb{X}$-Modal Reasoning

**Jiazuo Yu**[1], **Haomiao Xiong**[1], **Lu Zhang**[1,*], **Haiwen Diao**[1], **Yunzhi Zhuge**[1],
**Lanqing Hong**[2], **Dong Wang**[1], **Huchuan Lu**[1], **You He**[3], **Long Chen**[4]
[1]Dalian University of Technology, [2]Huawei Noah's Ark Lab
[3]Tsinghua University, [4]The Hong Kong University of Science and Technology
yujiazuo@mail.dlut.edu.cn, zhangluu@dlut.edu.cn

## Abstract

Multimodal Large Language Models (MLLMs) have gained significant attention due to their impressive capabilities in multimodal understanding. However, existing methods rely heavily on extensive modal-specific pretraining and joint-modal tuning, leading to significant computational burdens when expanding to new modalities. In this paper, we propose **PathWeave**, a flexible and scalable framework with modal-**path** s**w**itching and **expa**nsion abilities that enables MLLMs to continually **ev**olve on modalities for $\mathbb{X}$-modal reasoning. We leverage the concept of Continual Learning and develop an incremental training strategy atop pre-trained MLLMs, enabling their expansion to new modalities using uni-modal data, without executing joint-modal pretraining. In detail, a novel Adapter-in-Adapter (AnA) framework is introduced, in which uni-modal and cross-modal adapters are seamlessly integrated to facilitate efficient modality alignment and collaboration. Additionally, an MoE-based gating module is applied between two types of adapters to further enhance the multimodal interaction. To investigate the proposed method, we establish a challenging benchmark called **C**ontinual **L**earning of **M**odality (MCL), which consists of high-quality QA data from five distinct modalities: image, video, audio, depth and point cloud. Extensive experiments demonstrate the effectiveness of the proposed AnA framework on learning plasticity and memory stability during continual learning. Furthermore, PathWeave performs comparably to state-of-the-art MLLMs while concurrently reducing parameter training burdens by 98.73%. Our code locates at https://github.com/JiazuoYu/PathWeave.

## 1 Introduction

With recent advances in artificial intelligence, Large Language Models (LLMs) have demonstrated impressive capacities in language understanding and reasoning. The success of LLMs [1, 2, 3, 4] has spurred researchers to develop Multimodal LLMs (MLLMs) by integrating additional input for multimodal tasks, such as image-text understanding [5, 6, 7], audio recognition [8, 9] and 3D question answering [10, 11]. Aided by large-scale image-text paired data from the Internet [12, 13, 6, 14, 5], vision LLMs have become a thriving area in the research community. The typical framework comprises a visual encoder, a frozen or trainable LLM, and a projection module for vision-language alignment. Through stepwisely pretraining on large-scale image-text pairs and instruction tuning on specific datasets, vision LLMs exhibit promising generalization abilities on downstream applications such as detection [15], grounding [16, 17], and captioning [6, 14]. Subsequently, the LLM-based framework and training pipeline of vision LLMs serve as the basis and drive the extension to other modalities, including video [18, 19], audio [9, 8], and point cloud [11, 10]. However, these modal-

---

*Corresponding author.

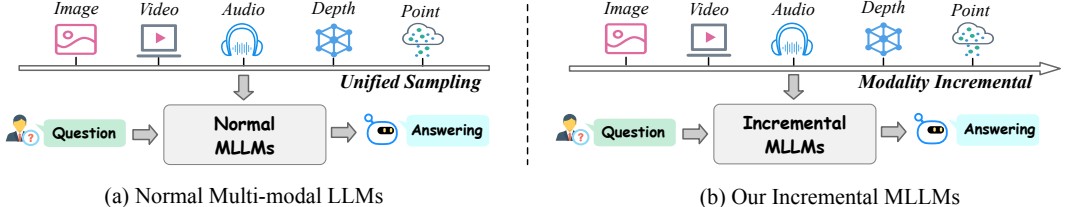

(a) Normal Multi-modal LLMs             (b) Our Incremental MLLMs

Figure 1: Comparisons of Different Multimodal LLMs: (a) The normal multimodal methods [21, 23, 22] require unified sampling across multi-modal. (b) Our proposed incremental MLLMs learns each modality sequentially without joint-modal datasets.

specific LLMs that inject single-modal data into language models struggle to tackle the challenge of perceiving different modalities like us humans.

To address this issue, recent approaches [20, 21, 22, 23] extend the architecture and training strategies of modal-specific MLLMs, and try to integrate multiple modalities into a unified system. Some early attempts [22, 23] utilize specific projection modules to align image, video, and audio encoders into a frozen LLM. However, a complex training process is usually required to enhance cross-modal alignment, involving separate pretraining on uni-modal data and joint fine-tuning on multimodal data. Subsequent attempts try to enhance the scalability of MLLMs by unifying the architecture and simplifying the training process. For instance, X-InstructBLIP [20] proposes a unified projection architecture for all modalities and constructs high-quality instruction tuning data to simplify modal-specific customization and pretraining. OneLLM [21] leverages a unified encoder and projection module and introduces an incremental pretraining strategy to achieve parameter unification for a wide range of modalities. While effective, most approaches still rely on joint-modal optimization that is high-resource demanding (see Figure 1 (a)). When expanded to new modalities, the models have to re-access all the historical data and repeat the complete training process, limiting the continual extension of MLLMs.

In this paper, we propose **PathWeave** ✑, a flexible and scalable framework with modal-**path** s**w**itching and **expa**nsion capabilities that enables MLLMs to continually **ev**olve on modality for $\mathbb{X}$-modal reasoning. PathWeave leverages the concept of Continual Learning (CL) and forms an incremental training pipeline on uni-modal data, eliminating the necessity for joint-modal pretraining or finetuning. To this end, we employ a pre-trained vision LLM [20] as the interface and propose a novel Adapter-in-Adapter (AnA) framework, allowing efficient extension and alignment for other modalities. We set two types of adapters in AnA, uni-modal and cross-modal, and seamlessly incorporate them to boost modality alignment and collaboration during incremental learning. Specifically, the uni-modal adapters are progressively added to the interface and optimized on the corresponding modality data, which will be frozen once trained. Meanwhile, we insert in-adapters into the previous uni-modal adapters to form cross-modal adapters, allowing the effective integration between historical knowledge and ongoing modality. Additionally, an MoE-based gating module is implemented between uni-modal and cross-modal adapters to further enhance multimodal collaboration. As shown in Figure 1 (b), our PathWeave can be flexibly implemented on the pretrained MLLMs and efficiently expand to more modalities in an incremental manner.

To evaluate the proposed PathWeave, we establish a challenging benchmark, namely **C**ontinual **L**earning of multi-**M**odality (**MCL**). It consists of data from five distinct modalities: image, video, depth, audio, and point cloud. In our setting, the modalities data are incrementally fed to the MLLMs. Thus, we leverage the commonly-used metrics from [20, 21] to investigate the precision on newly learned modalities. Furthermore, we introduce a metric to measure the forgetting rate in MCL to demonstrate the effectiveness of the proposed AnA strategy on historical modality memorization. Finally, we conduct extensive experiments to compare with state-of-the-art continual learning approaches, demonstrating that PathWeave is effective at incorporating multimodal data in an incremental manner. Moreover, our method achieves comparable performance with state-of-the-art MLLMs without requiring joint-modal pretraining or fine-tuning.

In summary, our contributions are summarized as follows:

• We present an efficient and scalable framework, PathWeave, which enables MLLM to progressively expand on multiple modalities, without the need for joint-modal pretraining.

- We introduce a novel adapter-in-adapter framework that seamlessly integrates uni-modal and cross-modal adapters to enhance modality alignment and interaction during incremental learning.
- We establish a challenging MCL benchmark with well-defined evaluation metrics. Extensive results demonstrate the effectiveness of PathWeave on modality plasticity and memorization during continual learning. Furthermore, PathWeave performs on par with state-of-the-art MLLMs while reducing parameter training burdens by at least 98.73%.

## 2 Related Work

**Multimodal Large Language Models.** In recent years, researchers have been exploring the potential of LLMs in multimodal perceptions, such as visual question answering [5, 14] and captioning [6, 24]. This leads to the rapid development of Multimodal LLMs [6, 5, 21, 22]. For example, LLaVA [5] utilizes a simple linear layer to project visual information into language space, enduing LLMs the ability to perceive natural scenes. Subsequently, several methods attempt to expand the supported modalities of LLMs by modifying architecture designs or training strategies. For instance, X-LLM [22] and Chatbridge [23] use modal-specific modules to extract features for multiple modalities and exploit modal-specific projection layers for multimodal alignment on a frozen LLM. However, a complex training process is usually required to enhance cross-modal alignment, which involves separate pretraining on uni-modal data and joint instruction tuning on multimodal data. Later, X-InstructBLIP [20] proposes a unified projection architecture (Q-former) for all modalities and collects large-scale, high-quality instruction tuning data to eliminate the need for uni-modal pretraining. OneLLM [21] explores parameter unification by introducing a unified encoder and projection module for a wide range of modalities. Although an incremental pretraining strategy is proposed to alleviate the high resource demand of cross-modal alignment, OneLLM still relies on cross-modal finetuning on large-scale instruction datasets. In contrast to these methods, we incorporate the continual learning concept into MLLMs and propose an incremental training strategy to allow MLLMs' modal expansion by finetuning on uni-modal data, without requiring joint-modal pretraining or finetuning. Among these approaches, X-InstructBLIP [20] is highly related to our method, as it separately tunes Q-former to align multimodal into a uniform system. However, our method designs an adapter-based expansible framework that significantly reduces the parameter training burdens by at least 98.73%.

**Continual Learning in Foundational Models.** Continual Learning (CL) has been applied to large foundational models [25, 26, 27, 28], allowing them to continually acquire new knowledge. To address the forgetting issue in CL, significant efforts [29] have been made, including data replay, regularization constraints, and dynamic frameworks. Data replay-based methods [30, 31, 32, 33, 34] retain the historical data in a memory bank and mix them with new data to execute the general training process. However, the redundant historical data would incur increasing resource demand during lifelong learning. Regularization-based methods add explicit regularization terms on weights [35, 36, 37] or data [38, 39, 40, 41] to achieve a balance between historical and new tasks, which are usually used as an auxiliary trick in data-replay or dynamic methods. In contrast, dynamic methods [28, 26, 42, 43, 44] exhibit impressive expansible abilities by incrementally adding new parameters into a shared interface. Recently, the dynamic frameworks have been combined with efficient tuning techniques to achieve efficient, cost-friendly continual learning on visual-textual domain [28, 27, 26]. This inspires us to eliminate joint-modal pertaining from MLLMs by developing an efficient, scalable framework where new modalities are incrementally involved by accessing uni-modal data. To this end, we propose an adapter-in-adapter framework, which incorporates uni-modal and cross-modal adapters for efficient modality alignment and collaboration.

**Transfer learning.** In the realm of Natural Language Progressing (NLP), fine-tuning large-scale models (*e.g.,* 175B GPT-3 [4]) imposes significant burdens in both parameter complexity and time consumption. As a result, transfer learning methods [45, 46, 47, 48] have gained significant attention to facilitate the efficient adaption of LLMs on downstream applications. The techniques usually activate a small set of parameters on the frozen models while achieving comparable performance with fully-finetuned approaches. Among these methods, LoRA [45] reduces the trainable parameters through low-rank matrix decomposition, leading to the generalization of the pre-trained model on diverse downstream tasks. The success of LoRA further promotes the development of parameter-efficient transfer learning of MLLMs [5, 49, 50] and uni-modal continual learning approaches [27, 26, 51]. However, these methods cannot be directly applied to fix the proposed MCL task due to the significant variations in modality spaces. In this paper, we propose a modality continual learning

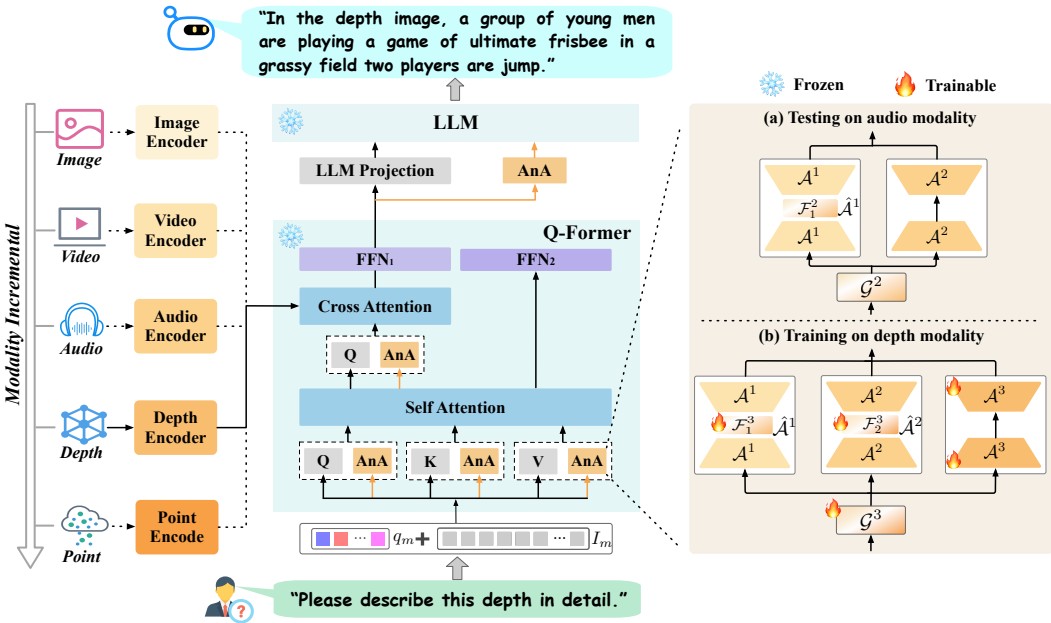

Figure 2: Overall framework of PathWeave. We start from a pretrained vision LLM [20] and progressively expand new modalities on it without acquiring historical data. Given input samples from **modality $m$**, we first exploit a frozen encoder ($E_m$) for feature extraction and leverage Q-Former to achieve multimodal alignment with LLMs. Then, the Adapter-in-Adapter (AnA) module is implemented in Q-Former to achieve flexible modal-path switching and expansion. In detail, the uni-modal adapters ($\mathcal{A}^m$) are implemented in parallel to facilitate new modal plasticity, which will be frozen once trained. While the cross-modal adapters ($\hat{\mathcal{A}}^m$) are formed by inserting a set of in-adapters ($\{\mathcal{F}_i^m\}_{i=1}^{m-1}$) into the learned uni-adapters to enhance the collaboration of historical knowledge. Additionally, an MoE-based gating module ($\mathcal{G}^m$) is implemented among uni-adapters to adaptively multimodal integration in input space.

method that incorporates adapter-based dynamic architecture on a frozen LLM, allowing efficient adaption and flexible expansion of new modalities in an incremental manner.

## 3 PathWeave

### 3.1 Preliminaries

Continual learning can empower large-scale foundation modals to constantly acquire new knowledge without accessing the entire historical data. We introduce this concept into MLLMs to form an incremental training strategy on uni-modal data called Continual Learning on Modality (MCL), eliminating the necessity of modal-specific pertaining and joint-modal datasets. Given a set of $M$ modalities $\{\mathcal{M}^m\}_{m=1}^M$, we enforce LLMs to sequentially access and learn on each modality for question answering. Here, each modality $\mathcal{M}^m$ contains $N^m$ datasets, which can be represented as $\mathcal{M}^m = \{\mathcal{D}_i^m\}_{i=1}^{N^m}$. More specifically, $\mathcal{D}_i^m = \{\boldsymbol{i}_i^m, \boldsymbol{s}_i^m, \boldsymbol{o}_i^m\}$ denotes the $i$-th data of the $m$-th modality $\mathcal{M}^m$, in which $\boldsymbol{i}, \boldsymbol{s}$ and $\boldsymbol{o}$ are text instruction, modality samples, and answering, respectively.

### 3.2 Framework Overview

This work presents PathWeave, an efficient and extensible framework that empowers MLLMs to constantly evolve on modalities, without requiring modal-specific pretraining. Considering the complicity of training MLLMs from scratch, we start from a pretrained vision LLM and align other modalities in an incremental manner. The overall framework of PathWeave is illustrated in Figure 2. Specifically, we build the PathWeave on X-InstructBLIP [20], providing a unified Q-Former architecture for various modalities. Given the samples from $m$-$th$ modality, a modal-specific encoder $E_m$ pretrained on the corresponding modality is first exploited for feature extraction. Then, the Q-Former $Q$ takes the input of modality feature, learnable query $q_m$, and instruction embedding $I_m$ for

multimodal alignment on a frozen LLM. It is worth noting that the initial modality $\mathcal{M}^0$ is predefined as images, as we leverage the pretrained X-InstructBLIP to facilitate the alignment of subsequent modalities. As a result, the entire parameter of the encoder, Q-Former, and LLM will be frozen during continual learning. To achieve continual learning on modalities, we propose Adapter-in-Adapter (AnA), a dynamically expansible framework atop MLLMs, enabling the efficient integration of new modalities by executing uni-modal instruction tuning. The AnA consists of uni-modal and cross-modal adapters to boost modality alignment and collaboration along the modality sequence. In detail, the uni-modal adapters ($\mathcal{A}^m$) are implemented in parallel in Q-Former to efficiently adapt to new modalities, which will be frozen once trained to "memorize" the historical modalities. Meanwhile, the cross-modal adapters ($\hat{\mathcal{A}}^m$) are constructed by inserting a set of in-adapters ($\{\mathcal{F}_i^m\}_{i=1}^{m-1}$) into previously learned uni-adapters to acquire their knowledge for ongoing modality, which will be removed accordingly when testing former modalities. Furthermore, an MoE-based gating module is implemented between uni-adapter and cross-adapted for further multimodal integration.

## 3.3 Adapter-in-Adapter

X-InstructBLIP [20] utilizes Q-Former as a unified framework to extend MLLMs' capabilities on more diverse modality reasoning, eliminating the need for modal-specific pretraining. However, instruction tuning on uni-modal data is implemented on separated Q-Formers, which leads to significant computational costs and parameter burdens when integrating more modalities. Recently, some attempts [26, 27] have demonstrated that adapters with few parameters can enhance the adaption of foundation modal on downstream tasks. Inspired by this, we leverage an effective transfer learning technique, LoRA [45], to serve as the basic unit of our AnA framework, enabling the efficient adaption of subsequent modalities during incremental learning.

**Uni-modal Adapters.** Given the current modality $\mathcal{M}^m$, we implement uni-modal adapters $\mathcal{A}^m$ in the pretrained Q-Former for new modal alignment. The adapters $\mathcal{A}^m$ are inserted into different linear layers $l$ of pretrained model in parallel. The output of layer $l$ with adapters $\mathcal{A}^m$ can be expressed as:

$$\boldsymbol{y}_l^m = Q_l(\boldsymbol{x}_l^m) + \mathcal{A}_l^m(\boldsymbol{x}_l^m), \tag{1}$$

where $\boldsymbol{x}_l^m$ and $\boldsymbol{y}_l^m$ are the input and output embedding of $l$-$th$ layer when aligning $m$-$th$ modality. $\mathcal{A}_l^m$ is the adapter of $m$-$th$ modality in $l$ layer, and $\mathcal{A}^m(\boldsymbol{x}) = \mathcal{F}_u^m(\mathcal{F}_d^m(\boldsymbol{x}))$, where $\mathcal{F}_u$ and $\mathcal{F}_d$ are the up and down projection of adapter. The uni-modal adapters are effective at acquiring modal-specific knowledge. Besides, the parallel architecture of adapters endows our system with the capabilities to flexibly switch and expand to diverse modalities.

**Cross-modal Adapters.** The uni-modal adapters are effective at preserving the uni-modal knowledge and alleviating the forgetting issue in long-term learning. Based on it, we introduce a modal-special in-adapter module ($\mathcal{F}_i^m$) to form a cross-modal adapter ($\hat{\mathcal{A}}^m$), which can help the ongoing modality learn previous knowledge and encourage inter-modality collaboration. Specifically, the in-adapters are inserted into the previously learned uni-modal adapters to effectively acquire the learned knowledge without reactivating their parameters. Then, the output of $l$-$th$ layer $\boldsymbol{y}_l^m$ after adding In-Adapter $\mathcal{F}_i^m$ can be reformulated as:

$$\boldsymbol{y}_l^m = Q_l(\boldsymbol{x}_l^m) + \sum_{i=1}^{m-1} \hat{\mathcal{A}}_l^i(\boldsymbol{x}_l^m) + \mathcal{A}_l^m(\boldsymbol{x}_l^m), \tag{2}$$

where $\hat{\mathcal{A}}^i(\boldsymbol{x}) = \mathcal{F}_u^i(\mathcal{F}_i^m(\mathcal{F}_d^i(\boldsymbol{x}))), i \in [1, m-1]$ represents the cross-modal adapters for current modality $\mathcal{M}^m$. $\mathcal{F}_i^m$ is the in-adapter that is inserted into $i$-$th$ frozen uni-adapters $\mathcal{A}^i$, which is a single linear layer with the dimension of adapters' low rank. The uni-modal and cross-modal adapters collaborate to facilitate the new modality alignment and cross-modal integration during incremental learning. Furthermore, the proposed in-adapter serves as a plug-and-play module that will not affect the performance of previously learned adapters, thereby effectively alleviating the modality forgetting.

**MoE-based Gating.** Cross-modal adapters rely on in-adapters to effectively leverage historical knowledge to boost the alignment of ongoing modality. However, the output of cross-modal and uni-modal adapters are treated equally in the original Q-Former. Considering the significant gap between distinct modalities, this simple integration strategy might pose performance degradation affected by the interfering information from other modalities. To address this issue, we propose an MoE-based gating module between cross-modal and uni-modal adapters for adaptive multimodal integration. Our MoE-based gating $\mathcal{G}^m$ automatically assigns weights of paths $\mathcal{P}^m$ of different

cross-modal adapters and uni-modal adapter to produce outcomes tailored to each modality $\mathcal{M}^m$. The paths $\{\mathcal{P}^m\}_{m=1}^M$ include the previous cross-modal adapters with the current in-adapter and current uni-modal adapter. Therefore, each linear's output $\boldsymbol{y}^m$ after adding MoE-based gating $\mathcal{G}^m$ in AnA module can be computed as:

$$\boldsymbol{y}_l^m = Q_l(\boldsymbol{x}_l^m) + \sum_{i=1}^m W_i^m \mathcal{P}_i(\boldsymbol{x}_l^m), \tag{3}$$

where $W^m = \{W_i^m\}_{i=1}^{N_E}$ represents the gating weights assigned by $\mathcal{G}^m$, dictating the contribution of each adapter's path $\mathcal{P}^m$. The gating weights are then computed as follows:

$$W^m = Softmax(\mathcal{G}^m(\boldsymbol{x}^m)), \tag{4}$$

where $\mathcal{G}^m$ projects each token of embeddings $\boldsymbol{x}$ to a 1-D vector indicating each modality's likelihood of functioning. It is worth noting that we do not set the $Topk$ hyper-parameter here. By default, the knowledge of each modality will provide a reference for the current modality. The $Softmax(\cdot)$ function normalizes these weights to emphasize the modality-branch contribution. Finally, the output $\boldsymbol{y}_l^m$ of AnA with MoE-based gating can be expressed as:

$$\boldsymbol{y}_l^m = Q_l^m(\boldsymbol{x}_l^m) + \sum_{i=1}^{m-1} W^i \hat{\mathcal{A}}^i(\boldsymbol{x}_l^m) + W^m \mathcal{A}^m(\boldsymbol{x}_l^m). \tag{5}$$

## 4  Continual Learning on Modality

**MCL Benchmark.** We establish a challenging benchmark, Continual Learning on Modality (MCL), which consists of multimodal high-quality QA data to evaluate the effectiveness of our method on continual uni-modal finetuning. These datasets are collected from five distinct modalities: image, video, depth, audio and point cloud. Based on this benchmark, our PathWeave is trained and tested along the multimodal sequence without requiring modal-specific pretraining or joint-modal finetuning. More details of the dataset list and size for each modality are illustrated in Table A6 of the Appendix.

**MCL Metrics.** We formulate the metrics from two aspects to evaluate the proposed MCL strategy on multimodal reasoning. On the one hand, we use the general metrics from MLLMs [20, 21] to investigate the model's overall performance on learned new modalities. On the other hand, we modify the conventional metrics of continual learning to verify the performance of our method on "catastrophic forgetting". Specifically, for each modality and dataset, suppose $S_{m,i}^n$ represents the evaluation score on $n$-th datasets of modality $\mathcal{M}^i$ after training on modality $\mathcal{M}^m$. We redefine the forgetting rate [28] to measure the degree of forgetting $F_m$ on all old modalities after each modality stage $m$:

$$F_m = \frac{1}{m} \sum_{i=0}^{m-1} F_{m,i}^{N_i}, \tag{6}$$

where $F_{m,i}^{N_i}$ is the average forgetting across $N_i$ datasets of modality $i$ after modality $m$ training, and $N_i$ is the number of datasets in modality $i$. And the $F_{m,i}^{N_i}$ are defined:

$$F_{m,i}^{N_i} = \frac{1}{N^i} \sum_{n=1}^{N_i} \max_{0 \le j < m} (S_{j,i}^n) - S_{m,i}^n. \tag{7}$$

In addition, we define the forgetting $\hat{F}_i^n$ for the $n$-th dataset in modality $i$ during the training of all modalities:

$$\hat{F}_i^n = \frac{1}{M-i} \sum_{m=i+1}^M \max_{0 \le j < m} (S_{j,i}^n) - S_{m,i}^n. \tag{8}$$

To measure the overall performance on learned modalities, we further report the average scores of across $N_m$ datasets of modality $m$ after training on $m$ modality, it can be expressed as:

$$T_m = \frac{1}{N_m} \sum_{n=1}^{N_m} S_{m,m}^n. \tag{9}$$

And the performance on learned modalities $\hat{T}_i^n$ for the $n$-th dataset in modality $i$ can be expressed as $\hat{T}_i^n = S_{i,i}^n$.

| Method | Image→Video | | Video→Audio | | Audio→Depth | | Depth→3D | |
|---|---|---|---|---|---|---|---|---|
| | $T_1 \uparrow$ | $F_1 \downarrow$ | $T_2 \uparrow$ | $F_2 \downarrow$ | $T_3 \uparrow$ | $F_3 \downarrow$ | $T_4 \uparrow$ | $F_4 \downarrow$ |
| Continual-FT | 51.33 | 25.50 | 60.97 | 57.74 | 93.55 | 68.19 | 149.9 | 65.34 |
| WiSE-FT[54] | 37.50 | 1.30 | 15.70 | 5.18 | 67.60 | 10.94 | 4.75 | 13.18 |
| L2 Reg&WE [25] | 39.05 | 0.60 | 7.33 | 0.05 | 70.00 | 4.27 | 6.75 | 4.45 |
| EProj[28] | **47.60** | **0.00** | 17.67 | **0.00** | 70.75 | **0.00** | 7.75 | **0.00** |
| Ours | 45.08 | **0.00** | **56.63** | **0.00** | **83.35** | **0.00** | **73.45** | **0.00** |

Table 1: Comparison with other CL methods on each modalities of in-domain datasets. We label the best and second methods with **bold** and underline styles. The top gray block indicates the upper-bound scores $T_m$ of transfer learning capability to adapt the new modality.

| | Method | COCO Val [55] | COCO Test [55] | MSRVTT [56] | MSRVTT QA [56] | AudioCaps Val [57] | AudioCaps Test [57] | AudioCaps QA [57] | CC3M [58] | LLAVA50K [5] | Cap3D QA [59] | Cap3D Cap [59] | *Average* |
|---|---|---|---|---|---|---|---|---|---|---|---|---|---|
| $\hat{T}_i^n \uparrow$ | Continual-FT | - | - | 59.4 | 43.3 | 62.4 | 74.7 | 45.8 | 104.4 | 82.7 | 41.7 | 108.2 | 69.20 |
| | WiSE-FT[54] | - | - | 40.5 | 34.5 | 9.5 | 10.5 | 27.1 | 84.9 | 50.3 | 4.2 | 5.3 | 29.64 |
| | L2 Reg&WE [25] | - | - | 43.8 | 34.3 | 14.4 | 3.4 | 4.2 | 87.4 | 52.6 | 3.2 | 10.3 | 28.20 |
| | EProj[28] | - | - | **55.1** | **40.1** | 17.7 | 10.0 | 25.3 | 86.1 | 55.4 | 4.9 | 10.6 | 33.91 |
| | Ours | - | - | 52.8 | 37.4 | 64.0 | 59.4 | 46.5 | 96.5 | 70.2 | 39.3 | 107.6 | **63.74**(+29.83) |
| $\hat{F}_i^n \downarrow$ | Continual-FT | 80.3 | 80.1 | 39.0 | 31.3 | 57.2 | 68.2 | 40.7 | 90.4 | 49.5 | - | - | 59.63 |
| | WiSE-FT[54] | 10.3 | 16.1 | 5.4 | 11.5 | 4.8 | 7.0 | 17.0 | 8.5 | 3.00 | - | - | 9.29 |
| | L2 Reg&WE [25] | 0.5 | **0.0** | 8.6 | 6.3 | 10.3 | 0.4 | **0.0** | 0.6 | **0.0** | - | - | 3.00 |
| | EProj[28] | **0.0** | **0.0** | **0.0** | **0.0** | **0.0** | **0.0** | **0.0** | **0.0** | **0.0** | - | - | **0.00** |
| | Ours | **0.0** | **0.0** | **0.0** | **0.0** | **0.0** | **0.0** | **0.0** | **0.0** | **0.0** | - | - | **0.00**(-3.00) |

Table 2: Comparison with other CL methods on the performance of each in-domain datasets. We label the best and second methods with **bold** and underline styles. The top gray block indicates the upper-bound scores $\hat{T}_i^n$ of transfer learning capability to adapt the new modality.

## 5 Experiments

### 5.1 Implementation Details

Our method is built on the LAVIS library's framework [52] atop the Vicuna v1.1 7b [3]. The input preprocessing method remains consistent with X-InstructBLIP [20]. We optimize our model on $4\times$A800 GPUs (80GB) using AdamW [53] with $\beta_1 = 0.9$, $\beta_2 = 0.999$, and a weight decay of 0.05. Our initial pre-trained model is the image modality model of X-InstructBLIP [20]. During training, the unified incremental module, consisting of Q-former and LLM projection, is continuously trained in the order of image, video, audio, depth, and 3D modalities. During testing, the learnable query and modality encoder are kept modality-specific. The CL methods compared below maintain consistent settings with our method. More details are provided in the Appendix A.2.

### 5.2 Comparison with State-of-the-art Methods

**Transfer Learning on New Modality.** As shown in Table 1 and 2, we conduct experiments on existing traditional CL methods under our proposed MCL setting. We report the average expansion capability for each modality, which is represented as $T_m$ and indicates the scalability in the new modality. The inference datasets are in-domain, which is involved in model training, and additional results of out-of-domain are provided in the supplementary material. Continual-FT, which refers to continuous learning of each modality without incorporating anti-forgetting strategies, exhibits the best expansion ability due to fine-tuning all parameters but inevitably leads to catastrophic forgetting. In contrast, the methods of L2 Reg&WE [25], WISE-FT [54] and EProj [28] effectively alleviate forgetting by parameter regularization and ensemble, but it is difficult for them to transfer new modality. As shown in Table 1, when performing transfer learning on new modalities with significant data distribution gaps from the images, these methods under-perform ours by at least 38 points on the Audio modality and 66 points on the 3D modality. Furthermore, as shown in Table 2, our method surpasses the current best methods by over 29 points in the average transfer learning metrics across in-domain datasets. This demonstrates that our approach can effectively prevent forgetting while flexibly extending to new modalities with substantial data distribution differences.

**Alleviate Forgetting of Previous Knowledge.** We also present the average forgetting rate $F_m$ of historical modality knowledge after training each modality $m$, as shown in the $F_m$ columns of Table 1

| Method | Params | All Modal | Data Size | Times† | GPU† | MSVD QA | Clotho Caps | Modelnet Cls |
|---|---|---|---|---|---|---|---|---|
| X-InstructBLIP [20] | 189.91M+ | ✗ | 27.78M+ | 0.34s/it | 28.7G | 51.7 | 29.4 | 62.8 |
| OneLLM [21] | 7B+ | ✓ | 1007M+ | 0.83s/it | 64.8G | 56.5 | 29.1 | - |
| X-LLM [22] | 189.91M+ | ✓ | 17.2M+ | 0.34s/it | 28.7G | - | - | - |
| ChatBridge [23] | 7B+ | ✓ | 4.4M+ | 0.34s/it | 28.7G | 45.3 | 26.2 | - |
| Ours | 0.8~2.4M | ✗ | 23.2M+ | 0.23s/it | 13.1G | 48.2 | 28.6 | 59.5 |

Table 3: Comparison with state-of-the-art methods on training parameters, data requirements and some performance. "All Modal" indicates whether fine-tuning on all modality datasets is included. "†" represents the same hyperparameters and training settings of different methods for fair comparison.

| Method | Video→Audio | | Audio→Depth | | Depth→3D | |
|---|---|---|---|---|---|---|
| | $T_{2(in)}$ ↑ | $T_{2(out)}$ ↑ | $T_{3(in)}$ ↑ | $T_{3(out)}$ ↑ | $T_{4(in)}$ ↑ | $T_{4(out)}$ ↑ |
| Continual-Adapter | 51.17 | 40.28 | 75.75 | 49.10 | 68.00 | 51.05 |
| w/o MoE-based gating | 43.77 | 39.35 | 76.40 | 49.80 | 69.50 | 49.70 |
| w/o In-Adapter | 52.47 | 40.78 | 79.50 | 50.25 | 71.35 | 52.60 |
| Ours | **56.63** | **42.90** | **83.35** | **52.20** | **73.45** | **53.70** |

Table 4: Ablation study of different parts for the influence of the each modalities' performance. We label the best and second methods with **bold** and underline styles.

and 2. The results show that continually full finetuning pre-trained modal suffers from catastrophic forgetting. WiSE-FT [54] and L2 Reg&WE [25] achieve some effectiveness in combating forgetting via parameter regularization and ensemble. However, the constraint of parameters limits their transfer learning on new modalities. In contrast, the EProj [28] and our method achieve anti-forgetting by freezing model parameters. However, the scalability of the EProj [28] is significantly lower than our method, especially in the audio and 3D modes. It indicates that our method achieves an optimal balance between anti-forgetting and effective expansion compared to other methods.

**Comparison with Existing MLLMs.** Table 3 shows the comparison between our approach and state-of-the-art multimodal QA methods in terms of training parameters, required data, training times, GPU usage, and relevant multimodal QA metrics. Among these methods, we unify the settings to ensure fairness in the Times and GPU metrics by only training on the instruction tuning stage, setting all batchsize to 4, and keeping the LLMs of BLIP-based X-LLM and ChatBridge frozen. It can be seen that our method demonstrates a significant advantage in parameter efficiency compared to X-InsructBLIP [20] and OneLLM [21], reducing parameter training burdens by at least 98.73%. Moreover, compared with OneLLM [21], X-LLM [22], and ChatBridge [23], our approach does not necessitate pre-training and instruction tuning with all joint-modal datasets to adapt to multimodal language reasoning tasks. Our method offers flexible scalability and requires considerably less training data than other methods. The results of the three QA tasks involving video, audio, and 3D, as shown in Table 3, indicate that our approach maintains flexibility without significantly compromising model performance. More experiments are provided in the Table A11 of Appendix.

## 5.3 Ablation Study

**Ablation Study of the In-Adapter and MoE-based gating.** We conduct detailed ablation studies on different parts of the proposed method, as shown in Table 4 and 5. Table 4 shows the average performance $T_m$ of transfer learning in each modality. It can be seen that our final method demonstrates increasingly significant performance improvements compared to others when faced with continual modality changes. For instance, as we further extend to depth and 3D modalities, the collaborative synergy between MoE-based gating and In-Adapter becomes increasingly apparent. In addition, Table 5 demonstrates that compared to directly using the incremental adapter method, our approach improves the average performance of transfer learning across all datasets by 4.3 points. When removing the In-Adapter or MoE-based gating, the model's transfer learning performance of transfer learning across all datasets decreases by at least 1.1 points and 4.0 points. It indicates the effectiveness of our proposed In-Adapter and MoE-based gating, which enhance inter-modal interactions and modulate cross-modal knowledge.

**Analysis of the Benefit from Previous Modalities.** Figure 3 presents the ablation study on the ability to transfer learning based on different knowledge of modalities. As shown in Figure 3 (a), our method enhances the scalability of audio modality after incorporating additional video modality training. It indicates that our designed method can extract knowledge from the other adapter to enhance the learning of the current modality. In addition, when more than one modality is additionally introduced, our method can still enhance new generalization by modulating inter-modal knowledge and fine-

| Method | AudioCaps Val [57] | AudioCaps Test [57] | AudioCaps QA [57] | ESC50 Cls [60] | ESC50 Open [60] | ClothoAQA [61] | Clotho Caps [62] | CC3M [58] | LLAVA50K [5] | NYU v2 [63] | SUN RGB-D [64] | Modelnet Cls [65] | Modelnet Open [65] | Cap3D QA [59] | Cap3D Cap [59] | *Average* |
|---|---|---|---|---|---|---|---|---|---|---|---|---|---|---|---|---|
| Continual-Adapter | 61.0 | **60.9** | 31.6 | 65.3 | 36.8 | 30.2 | **28.8** | 90.7 | 60.8 | 58.7 | 39.5 | 56.2 | 45.9 | 35.8 | 100.2 | 53.49 |
| w/o MoE-based gating | 52.1 | 51.0 | 28.2 | 67.1 | 34.1 | 28.7 | 27.5 | 92.5 | 60.3 | 58.4 | 41.2 | 55.8 | 43.6 | 36.1 | 102.9 | 51.97 |
| w/o In-Adapter | **71.4** | 58.3 | 27.7 | 69.5 | 34.7 | 31.2 | 27.0 | 93.9 | 65.1 | 59.1 | 41.4 | 58.5 | 46.7 | 37.9 | 104.8 | 55.15 |
| Ours | 64.0 | 59.4 | **46.5** | **72.6** | **36.9** | **33.5** | 28.6 | **96.5** | **70.2** | **62.2** | **42.2** | **59.5** | **47.9** | **39.3** | **107.6** | **57.79**(+2.64) |

Table 5: Ablation study of different parts for the influence of the each dataset's performance. We label the best and second methods with **bold** and underline styles.

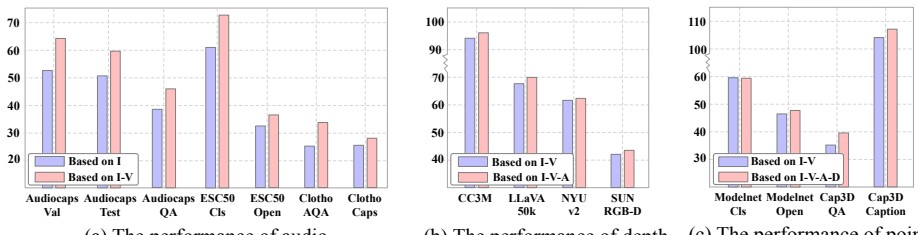

(a) The performance of audio    (b) The performance of depth    (c) The performance of point

Figure 3: Ablation study of the $\hat{T}_i^n$ performance for the $n$-$th$ dataset in modality $i$, which benefits from knowledge of different modalities. "Based on I-V-A-D" represents training point modality based on our pre-trained PathWeave that is trained in the sequence of image, video, audio, and depth.

tuning frozen knowledge with In-Adapter, as shown in Figure 3 (b) and (c). It demonstrates that our method can enhance the adapting to new modalities by knowledge learned from other modalities.

## 5.4 Qualitative Analysis

Figure 4 shows the qualitative results of our method for inference on each modality after continual training is completed. We show our final model can (a) understand visual content in images, (b) leverage temporal information in videos, (c) scene understanding using depth maps, (d) do creative writing based on audio content, and (e) understand the details of 3D shapes. More qualitative results are provided in Figure A5 of the Appendix.

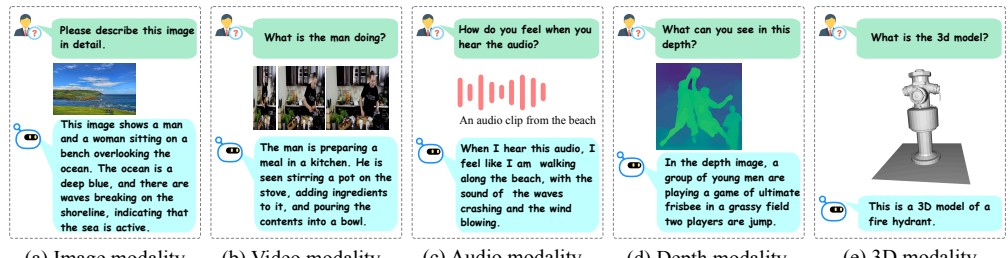

(a) Image modality    (b) Video modality    (c) Audio modality    (d) Depth modality    (e) 3D modality

Figure 4: Qualitative results of our method on each modality after continuous training.

## 6 Conclusion and Discussion

We propose a flexible and scalable framework for multi-modal language reasoning that enables MLLMs to continually expand on multiple modalities without joint-modal datasets. We introduce an incremental Adapter-in-Adapter (AnA) strategy, incorporating two types of adapters to enhance modality plasticity and collaboration during expanding on other modalities. Moreover, we design an MoE-based gating module to further enhance multi-modal integration by modulating the output space of different modalities. Extensive experimental results in our proposed benchmark demonstrate the superiority of our method over previous arts in terms of modality alignment and memorization.

A limitation of this paper is that we only explored the extension of five modalities and do not cover all modal information in real-world scenarios. Furthermore, the implicit interaction between the modalities in our method cannot accomplish cross-modal joint language reasoning tasks in an incremental manner.

## Acknowledgements

This work was supported by National Natural Science Foundation of China under Grant 62206039, 62293544, and the Fundamental Research Funds for the Central Universities (DUT24RC(3)025).

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

# A  Appendix

## A.1   Dataset Details

We summarize the multimodal-text dataset in Table A6 for modality continue learning. For depth-text pairs, we adopt the DPT model pre-trained on ominidata [66] to generate depth maps. The source dataset is a subset of CC3M [58], around 0.5M image-text pairs and 50K image-text pairs random sampled from LLaVA-150K [5].

LLaVA data includes multiple rounds of dialogue. To align with our training process, we randomly select one round as a training sample. This selection method also applies when creating the validation set, where these samples remain fixed and do not change during testing.

| Modality | Size | Dataset |
|---|---|---|
| Image | 21.3M | MS COCO [67], CapFilt14M [6], CC12M [55] SBU Captions [68], Visual Genome [69], AOK VQA [70] OK VQA [71], OCR VQA [72], Visual Genome QA [69] VQAV2 [73], LLaVA150K [5] |
| Video | 0.2M | MSRVTT [56], MSRVTT-QA [74] |
| Audio | 0.3M | WavCaps [75], AudioCaps [57], AudioCaps-QA [57] |
| 3D | 0.9M | Cap3D [59], Cap3D-QA |
| Depth* | 0.5M | CC3M [58], LLaVA-50K [5] |
| **Total** | **23.2M+** | All Datasets |

Table A6: Datasets for continually uni-modal finetuning. Our datasets are extensions of X-InstructBLIP [20], in contrast, we additionally included depth data and removed inaccessible video data WebVid2M [76]. * represent data we generated ourselves.

## A.2   Training & Evaluation Details

Table A7 records the detailed hyper-parameters we used during the training and testing process. It is worth noting that the training of our method on each modal data is continuous. The encoders for image, video, and depth are set to EVA-CLIP-ViT-G/14 [77]. The audio and 3D encoders are $BEATs_{iter3+}$ [78] and ULIP-2, respectively.

When using WiSE-FT [54] and L2 Reg&WE [25] methods for training, in order to be as consistent as possible with the original approach, we update the weights of the Q-Former and LLM projection layer in each inner epoch (for WiSE-FT, we set update coefficient $\alpha$ as 0.8). For example, when we train on Audio modal data, the total training iteration is set to 65000, and 5000 iterations per inner epoch, then the number of weight updates is 13 times in the current situation.

During modality backward testing for methods in Table 1, we keep the encoder and Q-Former queries consistent with the test modality. We utilize the same instruct prompts as X-InstructBLIP [20] during training and testing.

| Modality | Iteration | Batch Size (Train/Val) | Learning Rate |
|---|---|---|---|
| Video | 15K | 16/8 | 1e-5 |
| Audio | 65K | 16/8 | 1e-5 |
| Depth | 35K | 4/8 | 1e-5 |
| 3D | 65K | 16/16 | 1e-5 |

Table A7: Hyper-parameters for modality continue learning. We keep all the learning rate decrease from 1e-5 and cosine annealing strategy with 0.5 decay weight. The warm-up phase starts from 1e-8 and lasts for 1000 iterations for all modality training.

## A.3   Complete Raw Data

Table A8 records all the original data of the methods compared in Table 1. We highlight the transfer learning performance in new modality of each method with green color.

Table A8: Raw data records of all compared CL methods in all modalities.

| Method | Modality | Image modality | | | Video modality | | | | Audio modality | | | | | | | Depth modality | | | | Point modality | | | |
|---|---|---|---|---|---|---|---|---|---|---|---|---|---|---|---|---|---|---|---|---|---|---|---|
| | | GQA [79] | COCO Val [55] | COCO Test [55] | MSVD QA [74] | MSVD Cap [80] | MSRVTT [56] | MSRVTT QA [56] | AudioCaps Val [57] | AudioCaps Test [57] | AudioCaps QA [57] | ESC50 Cls [60] | ESC50 Open [60] | ClothoAQA [61] | Clotho Caps [62] | CC3M [58] | LLAVA50K [5] | NYU v2 [63] | SUN RGB-D [64] | Modelnet Cls [65] | Modelnet Open [65] | Cap3D QA [59] | Cap3D Cap [59] |
| Ours | Image | 48.1 | 137.7 | 138.2 | - | - | - | - | - | - | - | - | - | - | - | - | - | - | - | - | - | - | - |
| | Video | 48.1 | 137.7 | 138.2 | 48.2 | 106.9 | 52.8 | 37.4 | - | - | - | - | - | - | - | - | - | - | - | - | - | - | - |
| | Audio | 48.1 | 137.7 | 138.2 | 48.2 | 106.9 | 52.8 | 37.4 | 52.1 | 51.0 | 28.2 | 67.1 | 34.1 | 28.7 | 27.5 | - | - | - | - | - | - | - | - |
| | Depth | 48.1 | 137.7 | 138.2 | 48.2 | 106.9 | 52.8 | 37.4 | 52.1 | 51.0 | 28.2 | 67.1 | 34.1 | 28.7 | 27.5 | 92.5 | 60.3 | 58.4 | 41.2 | - | - | - | - |
| | Point | 48.1 | 137.7 | 138.2 | 48.2 | 106.9 | 52.8 | 37.4 | 52.1 | 51.0 | 28.2 | 67.1 | 34.1 | 28.7 | 27.5 | 92.5 | 60.3 | 58.4 | 41.2 | 55.8 | 43.6 | 36.1 | 102.9 |
| EProj | Image | 48.1 | 137.7 | 138.2 | - | - | - | - | - | - | - | - | - | - | - | - | - | - | - | - | - | - | - |
| | Video | 48.1 | 137.7 | 138.2 | 48.7 | 125.6 | 55.1 | 40.1 | - | - | - | - | - | - | - | - | - | - | - | - | - | - | - |
| | Audio | 48.1 | 137.7 | 138.2 | 48.7 | 125.6 | 55.1 | 40.1 | 17.7 | 10.0 | 25.3 | 12.9 | 34.1 | 18.6 | 9.3 | - | - | - | - | - | - | - | - |
| | Depth | 48.1 | 137.7 | 138.2 | 48.7 | 125.6 | 55.1 | 40.1 | 17.7 | 10.0 | 25.3 | 12.9 | 34.1 | 18.6 | 9.3 | 92.5 | 60.3 | 61.9 | 40.7 | - | - | - | - |
| | Point | 48.1 | 137.7 | 138.2 | 48.7 | 125.6 | 55.1 | 40.1 | 17.7 | 10.0 | 25.3 | 12.9 | 34.1 | 18.6 | 9.3 | 86.1 | 55.4 | 61.9 | 40.7 | 15.3 | 13.2 | 4.9 | 10.6 |
| L2 Reg + WE | Image | 48.1 | 137.7 | 138.2 | - | - | - | - | - | - | - | - | - | - | - | - | - | - | - | - | - | - | - |
| | Video | 47.1 | 136.4 | 138.1 | 47.1 | 100.4 | 55.1 | 34.3 | - | - | - | - | - | - | - | - | - | - | - | - | - | - | - |
| | Audio | 45.8 | 131.2 | 138.5 | 45.1 | 105.1 | 34.1 | 28.5 | 14.4 | 3.4 | 4.2 | 2.0 | 0.8 | 13.4 | 1.6 | - | - | - | - | - | - | - | - |
| | Depth | 43.7 | 127.8 | 116.2 | 42.1 | 87.5 | 35.3 | 29.3 | 3.9 | 3.3 | 12.1 | 1.6 | 0.1 | 12.5 | 1.7 | 87.4 | 52.6 | 61.9 | 43.4 | - | - | - | - |
| | Point | 40.5 | 114.3 | 104.7 | 36.5 | 69.3 | 36.1 | 31.0 | 4.3 | 2.8 | 13.0 | 1.7 | 0.1 | 12.1 | 1.6 | 86.8 | 55.6 | 60.4 | 43.7 | 1.8 | 0.5 | 3.2 | 10.3 |
| WiSE-FT | Image | 48.1 | 137.7 | 138.2 | - | - | - | - | - | - | - | - | - | - | - | - | - | - | - | - | - | - | - |
| | Video | 47.7 | 136.6 | 138.1 | 47.1 | 82.7 | 40.5 | 34.5 | - | - | - | - | - | - | - | - | - | - | - | - | - | - | - |
| | Audio | 45.8 | 131.2 | 130.8 | 45.7 | 94.2 | 43.1 | 28.5 | 9.5 | 10.5 | 27.1 | 5.2 | 2.1 | 14.4 | 1.8 | - | - | - | - | - | - | - | - |
| | Depth | 42.3 | 116.2 | 116.2 | 43.7 | 74.2 | 34.6 | 20.4 | 5.8 | 4.5 | 13.2 | 2.8 | 0.3 | 10.5 | 1.7 | 84.9 | 50.3 | 60.7 | 39.9 | - | - | - | - |
| | Point | 40.5 | 114.3 | 104.7 | 36.5 | 69.3 | 30.2 | 20.2 | 3.7 | 2.6 | 7.1 | 3.5 | 0.1 | 4.3 | 0.8 | 76.4 | 47.3 | 52.8 | 32.7 | 13.5 | 8.5 | 4.2 | 5.3 |
| Continual FT | Image | 48.1 | 137.7 | 138.2 | - | - | - | - | - | - | - | - | - | - | - | - | - | - | - | - | - | - | - |
| | Video | 31.8 | 136.6 | 136.8 | 50.7 | 136.5 | 59.4 | 43.2 | - | - | - | - | - | - | - | - | - | - | - | - | - | - | - |
| | Audio | 40.7 | 59.5 | 61.0 | 39.7 | 45.7 | 16.2 | 10.9 | 62.4 | 74.7 | 45.8 | 66.2 | 18.4 | 24.3 | 26.1 | - | - | - | - | - | - | - | - |
| | Depth | 1.2 | 19.5 | 20.8 | 0.6 | 35.8 | 16.2 | 16.2 | 7.2 | 10.4 | 1.5 | 35.1 | 17.8 | 0.0 | 7.9 | 104.4 | 82.7 | 62.1 | 44.6 | - | - | - | - |
| | Point | 27.5 | 37.8 | 38.4 | 18.1 | 35.8 | 22.9 | 8.9 | 3.2 | 2.6 | 8.7 | 3.5 | 4.7 | 7.2 | 1.7 | 14.0 | 33.2 | 44.8 | 31.3 | 62.5 | 50.4 | 41.7 | 108.2 |

## A.4 Additional Experiments

As shown in Table A9 and Table A10, we conduct experiments to analyze the performance on out-of-domain data in addition to the in-domain experiments. Our method shows robust generalization while maintaining anti-forgetting performance on out-of-domain data. Specifically, compared with the full-finetune method, our average accuracy only decrease 0.33 points, while achieving 31.34 points anti-forgetting capability. At the same time, with the same powerful anti-forgetting ability as EProj [28], the generalization of our method between different modalities improves 18.95 points.

| Method | Image→Video | | Video→Audio | | Audio→Depth | | Depth→3D | |
|---|---|---|---|---|---|---|---|---|
| | $T_1 \uparrow$ | $F_1 \downarrow$ | $T_2 \uparrow$ | $F_2 \downarrow$ | $T_3 \uparrow$ | $F_3 \downarrow$ | $T_4 \uparrow$ | $F_4 \downarrow$ |
| Continual-FT | 93.60 | 16.30 | 33.75 | 34.63 | 53.35 | 45.30 | 56.45 | 49.67 |
| WiSE-FT[54] | 69.95 | 1.00 | 5.88 | **0.00** | 50.15 | 4.60 | 11.00 | 7.87 |
| L2 Reg&WE [25] | 73.75 | 0.40 | 4.45 | **0.00** | **52.65** | 3.99 | 1.15 | 5.20 |
| EProj[28] | **87.15** | **0.00** | 10.9 | **0.00** | 51.3 | **0.00** | 14.25 | **0.00** |
| Ours | 77.54 | **0.00** | **42.9** | **0.00** | 52.2 | **0.00** | **53.7** | **0.00** |

Table A9: Comparison with other CL methods on each modality of out-of-domain datasets. We label the best and second methods with **bold** and underline styles. The top block indicates the upper-bound scores $T_m$ of transfer learning capability to adapt the new modality.

| | Method | GQA [79] | MSVD QA [74] | MSVD Cap [80] | ESC50 Cls [60] | ESC50 Open [60] | ClothoAQA [61] | Clotho Caps [62] | NYU v2 [63] | SUN RGB-D [64] | Modelnet Cls [65] | Modelnet Open [65] | *Average* |
|---|---|---|---|---|---|---|---|---|---|---|---|---|---|
| $\hat{T}_i^n \uparrow$ | Continual-FT | - | 50.7 | 136.5 | 66.2 | 18.4 | 24.3 | 26.1 | 62.1 | 44.6 | 62.5 | 50.4 | 54.18 |
| | WiSE-FT[54] | - | 45.7 | 94.2 | 5.2 | 2.1 | 14.4 | 1.8 | 60.7 | 39.6 | 13.5 | 8.5 | 28.57 |
| | L2 Reg&WE [25] | - | 47.1 | 100.4 | 2.0 | 0.8 | 13.4 | 1.6 | 61.9 | **43.4** | 1.8 | 0.5 | 27.29 |
| | EProj[28] | - | **48.7** | **125.6** | 12.9 | 2.8 | 18.6 | 9.3 | 61.9 | 40.7 | 15.3 | 13.2 | 34.90 |
| | Ours | - | 48.2 | 106.9 | **72.6** | **36.9** | **33.5** | **28.6** | **62.2** | 42.2 | **59.5** | **47.9** | **53.85**(+18.95) |
| $\hat{F}_i^n \downarrow$ | Continual-FT | 22.8 | 36.5 | 96.1 | 46.9 | 7.15 | 20.7 | 21.3 | 17.3 | 13.3 | - | - | 31.34 |
| | WiSE-FT[54] | 6.1 | 5.1 | 33.4 | 3.6 | 1.9 | 7.0 | 0.6 | 7.9 | 6.9 | - | - | 8.05 |
| | L2 Reg&WE [25] | 0.2 | 3.1 | 8.7 | 0.4 | 0.7 | 1.1 | **0.0** | 0.5 | **0.0** | - | - | 1.62 |
| | EProj[28] | **0.0** | **0.0** | **0.0** | **0.0** | **0.0** | **0.0** | **0.0** | **0.0** | **0.0** | - | - | **0.00** |
| | Ours | **0.0** | **0.0** | **0.0** | **0.0** | **0.0** | **0.0** | **0.0** | **0.0** | **0.0** | - | - | **0.00**(-1.62) |

Table A10: Comparison with other CL methods on the performance of each out-of-domain dataset. We label the best and second methods with **bold** and underline styles. The top block indicates the upper-bound scores $\hat{T}_i^n$ of transfer learning capability to adapt the new modality.

We quantitatively compare the results of our method and other multi-modal large language models that support multiple modalities in Table A11. Compared with other MLLMs, we achieve a better trade-off between model performance and the number of supported modalities with fewer learnable parameters and less training data.

| Method | GQA | MSVD | MSVD Cap | ESC50 Cls | ESC50 Open | ClothoAQA | Clotho Caps | NYU v2 | SUN | Modelnet | Modelnet Open |
|---|---|---|---|---|---|---|---|---|---|---|---|
| X-InstructBLIP [20] | 48.1 | 52.5 | 118.2 | 75.9 | 38.2 | 15.4 | 29.4 | - | - | 62.8 | 46.7 |
| OneLLM [21] | 59.5 | 56.8 | - | - | - | 57.9 | 29.1 | 50.9 | 29.0 | - | - |
| ChatBridge [23] | - | - | - | 45.3 | 26.2 | - | - | - | - | - | - |
| Ours | 47.8 | 48.2 | 106.9 | 72.6 | 36.9 | 33.5 | 28.6 | 62.2 | 42.2 | 59.5 | 47.9 |

| Method | COCO$_{val}$ | COCO$_{test}$ | MSRVTT | MSRVTTQA | AudioCaps$_{val}$ | AudioCaps$_{test}$ | AudioCapsQA | CC3M | LLaVA | Cap3D QA | Caps3D Cap |
|---|---|---|---|---|---|---|---|---|---|---|---|
| X-InstructBLIP [20] | 137.7 | 138.2 | 58.8 | 41.3 | 62.7 | 58.3 | 37.4 | - | - | 48.0 | 134.1 |
| OneLLM [21] | - | - | - | 56.5 | - | - | - | - | - | - | - |
| ChatBridge [23] | - | - | - | 45.3 | 26.2 | - | - | - | - | - | - |
| Ours | 137.8 | 138.7 | 52.8 | 37.4 | 64.0 | 59.4 | 46.5 | 96.5 | 70.2 | 39.3 | 107.6 |

Table A11: Comparison with state-of-the-art methods on metrics of different datasets.

In addition, we provide more qualitative results on each modality in Figure A5.

## A.5 More training details

All modalities are trained by an Autoregressive CE loss. The detailed hyperparameter settings for each modality are shown in Table A12 of the attached PDF. We will provide further details and descriptions of the loss and hyperparameters in the paper to ensure better clarity and flow.

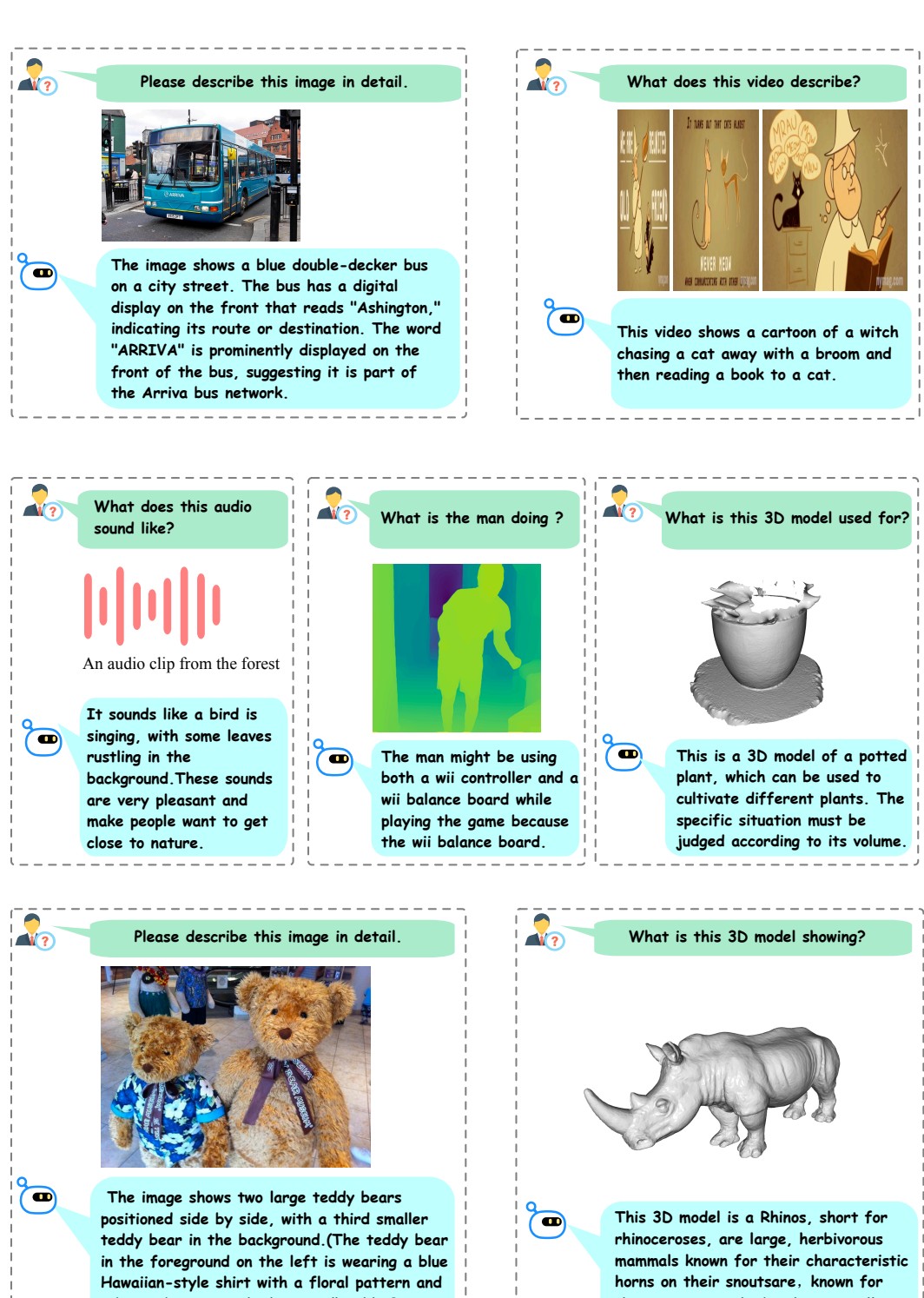

Figure A5: More qualitative results of our method on each modality after continuous training.

| Modality | Dataset | Prompt | Len. Penalty | Min Len. | Max Len. |
|---|---|---|---|---|---|
| Image | GQA [79] | based on the given the image respond to {} | -1. | 1 | 10 |
| | COCO Val [55] | a short description | 1. | 10 | 80 |
| | COCO Test [55] | a short description | 1. | 10 | 80 |
| Video | MSVD QA [74] | based on the given video respond to {} | -1. | 1 | 10 |
| | MSVD Cap [80] | a short description | 1. | 10 | 80 |
| | MSRVTT [56] | a short description | 1. | 10 | 80 |
| | MSRVTT QA [56] | based on the given video respond to {} | -1. | 1 | 10 |
| Audio | AudioCaps Val [57] | a short description. | 1. | 10 | 80 |
| | AudioCaps Test [57] | a short description. | 1. | 10 | 80 |
| | AudioCaps QA [57] | Question: {} Answer: | -1. | 1 | 10 |
| | ESC50 Cls [60] | describe the audio. | 1. | 1 | 80 |
| | ESC50 Open [60] | describe the audio. | 1. | 10 | 80 |
| | ClothoAQA [61] | Question: {} Answer: | -1. | 1 | 10 |
| | Clotho Caps [62] | a short description. | 1. | 10 | 80 |
| Depth | CC3M [58] | A short description of the depth: | 1. | 8 | 30 |
| | LLAVA50K [5] | Question: {} Answer: | 1. | 8 | 30 |
| | NYU v2 [63] | {class} What is the category of this scene? Choice one class from the class sets. | 1. | 8 | 30 |
| | SUN RGB-D [64] | {class} What is the category of this scene? Choice one class from the class sets. | 1. | 8 | 30 |
| Point | Modelnet Cls [65] | describe the 3d model. | 0. | 10 | 80 |
| | Modelnet Open [65] | based on the given input respond to {}. | 0. | 1 | 80 |
| | Cap3D QA [59] | describe the 3d model. | 1. | 1 | 3 |
| | Cap3D Cap [59] | describe the 3d model. | 1. | 1 | 3 |

Table A12: More details of hyperparameters used on each of the datasets.

