# OpenReview forum: "LLMs Can Evolve Continually on Modality for $\mathbb{X}$-Modal Reasoning"
_NeurIPS.cc/2024/Conference — NeurIPS 2024 poster_

### Official Review · Reviewer_fRFW · 2024-07-11

**Soundness:** 4
**Presentation:** 3
**Contribution:** 3
**Rating:** 7
**Confidence:** 4

**Summary:**

The authors introduce continual learning into MLLMs to explore the ability of pre-trained LLMs to evolve continually on multiple modalities while keeping knowledge from being forgotten. A novel PathWeave with Adapter-in-Adapter (AnA) is proposed, in which uni-modal and cross-modal adapters are seamlessly integrated to facilitate efficient modality alignment and collaboration. The authors establish a challenging benchmark, MCL, to investigate the proposed method’s performance on the new modality and all previous knowledge. The experimental results are encouraging, significantly reducing training parameter burdens.

**Strengths:**

1. The paper is logical, fluent, and easy to understand.

2. The proposed method enables existing pre-trained large models to progressively expand on multiple modalities without requiring joint training on all modalities. This idea of continually learning knowledge from pre-trained models to expand the modality is novel and could inspire the further exploration of multimodal works.

3. This paper establishes a challenging MCL benchmark to explore the generalization and anti-forgetting of cross-modal continual learning of pre-trained models. It is a promising benchmark that evaluates the performance of modality expansion of pre-trained MLLMs.

4. This paper conducted sufficient experiments on five modalities and more than 20 datasets. The performance is comparable to the joint training MLLMs while significantly reducing parameter burden. In addition, the comparison with other continual learning methods shows the state-of-the-art generalization performance. More experiments in the supplementary materials further demonstrate the effectiveness of this method.

**Weaknesses:**

1. Compared with fine-tuning all parameters, the performance of the adapter fine-tuning method still has room for improvement. I think some necessary discussion, analysis, or experimentation should be conducted.

2. What are the similarities and differences between this method and VPGTrans[1]? It is a highly efficient Visual Prompt Generator Transfer across LLMs with less training data and even task improvements. I believe that a related analysis of these two works is needed.

3. Some minor issues. 1) It seems that the modality sequence in Fig. 2 is unmatched with the experiments in Table 1. 2) In Fig. 2, the upper and lower trapezoids of A1, A2, etc. of AnA are represented by the same symbol. Do they share the parameters? 3) Some training details, including loss functions and hyperparameters for each modality, are not clear.

[1] Vpgtrans: Transfer visual prompt generator across llms. NeurIPS 2023.

**Questions:**

See weakness.

---

> ### Author Rebuttal · Authors · 2024-08-06
>
> - **Weakness 1: The performance gap between the PEFT and fully finetuning method.**
>
>     $\quad$ Thanks for your comments. Our method does fall slightly behind in performance metrics compared to fully fine-tuning approaches. This is primarily because that the Parameter-Efficient Fine-Tuning (PEFT) method naturally affects performance compared to fully fine-tune all parameters, as shown in the Continual-Adapter of Tables 4 and 5 of the main paper. **More importantly**, the main advantage of our method is the ability to flexibly and efficiently extend to multiple modalities sequentially while reducing the forgetting of previous modalities. Specifically,
>
>   - Our method reduces the number of trainable parameters by 98.73\% compared to fully fine-tuning methods.
>   - In contrast, our method demonstrates significantly superior anti-forgetting capabilities compared to fully fine-tuning methods.
>   - Compared to other anti-forgetting methods, our approach achieves state-of-the-art performance on transfer learning metrics $T_n$ and $\hat{T}_i^n$ and anti-forgetting $F_n$ and $\hat{F}_i^n$, as shown in Tables 1 and 2 of the main paper.
>
>   $\quad$ Therefore, these comparisons and results demonstrate that our method achieves the optimal balance between flexible modality extension and anti-forgetting. To further highlight our contribution, we have added related discussion and analysis in the revised version.
>
> - **Weakness 2: Related analysis between this paper and VPGTrans [1].**
>
>   $\quad$ Thank you for bringing this related work to our attention. We summarize the differences and similarities between the related work and ours as follows:
>
>   $\quad$ **Similarities**:
>
>   - Both works emphasize leveraging previous knowledge to facilitate new transfer tasks.
>   - Both works have studied the transfer learning performance based on the BLIPs architecture, with LLMs kept frozen.
>
>   $\quad$ **Differences**:
>   - Our work focuses on transfer learning across different modality datasets, while VPGTrans focuses on transfer learning across different language models.
>   - We address the issue of forgetting for previous modalities by continual learning strategy, whereas VPGTrans does not consider the problem of forgetting for previous models.
>   - VPGTrans has utilized the warming-up process on the projection layer to prevent performance drop and expedite VPG training.
>
>   $\quad$ The similarities between the two works indicate that transfer learning for new modalities or tasks can benefit from existing modalities or models. The comparison with VPGTrans can highlight our advantages in modality expansion and alleviate forgetting of previous knowledge, while also inspiring us to further optimize transfer learning performance in new modalities through warming-up strategies on some network layers. We will add related analysis and discussion in the revised version.
>
> - **Weakness 3: Some minor issues and more hyperparameters.**
>
>   $\quad$ Thanks for your comments and reminders. we have revised and added related content in the following three aspects.
>
>   - **Modality order correction.** Thanks for your reminder. we have corrected the order of modality in Figures 1 and 2 of the main paper.
>
>   - **Further descriptions of the notation in adapters.** The upper and lower trapezoids of A1, A2, etc. of AnA represent the up linear projection and down linear projection of LoRA, whose parameters are not shared. The relevant descriptions are on lines 181 and 182 of the main paper. We will add the above parameter settings in the revised version.
>
>   - **More training details.** All modalities are trained by an Autoregressive CE loss. The detailed hyperparameter settings for each modality are shown in Table R\#fRFW-1 of the attached PDF. We will provide further details and descriptions of the loss and hyperparameters in the paper to ensure better clarity and flow.
>
> >  [1] Vpgtrans: Transfer visual prompt generator across llms. NeurIPS 2023.

---

> > ### Comment · Reviewer_fRFW · 2024-08-11
> >
> > The rebuttal has addressed my initial concerns. Overall, I believe the proposed contributions are clearly demonstrated, and the paper is well-organized. Therefore, I will keep my rating as acceptance.

---

> > > ### Author Response · Authors · 2024-08-11
> > >
> > > Thank you for reading the response and your support of our work! We are glad that we have addressed your concerns.

---

### Official Review · Reviewer_odey · 2024-07-13

**Soundness:** 3
**Presentation:** 3
**Contribution:** 3
**Rating:** 5
**Confidence:** 1

**Summary:**

Due to a serious illness I have been experiencing recently, I deeply regret to inform you that I am unable to complete the review as scheduled. I kindly request the Chair to consider the opinions of the other reviewers

**Strengths:**

-

**Weaknesses:**

-

**Questions:**

-

**Limitations:**

-

---

### Official Review · Reviewer_PRBt · 2024-07-14

**Soundness:** 3
**Presentation:** 4
**Contribution:** 3
**Rating:** 6
**Confidence:** 5

**Summary:**

This paper proposes a flexible and scalable framework, PathWeave, which enables MLLMs to allow MLLMs gradually to involve reasoning ability on diverse modalities. The introduction of the adapter-in-adapter structure effectively alleviates the heavy burdens of the joint training or data replay strategies in previous MLLM methods. Extensive experiments demonstrate that the proposed method achieves state-of-the-art performance while significantly reducing training parameters by 98.73% compared to OneLLM and X-InstructBLIP.

**Strengths:**

This paper proposes an interesting framework to remedy the parameter and data burdens existed in training MLLMs. The proposed adapter-in-adapter framework exists novelties in multi-modal interaction and single-modal learning. Besides, they also provide a new benchmark to support the continual training and evaluation.
+Contribution&Results: This paper proposes a novel PathWeave, which combines the transfer learning and continual learning method to progressively expand LLMs on multiple modalities. And the paper proposes a new benchmark MCL to evaluate the model’s overall performance on learned modalities and the performance on “catastrophic forgetting”. The extensive experimental results provide sufficient experimental details and verification, which validates the effectiveness and superiority of the proposed methods.

+Inspiration: The proposed method exhibits promising experimental results for inspiring the subsequent work. Specifically, Figure 3 shows the positive effect of different modalities’ knowledge on special modal expansion, which is promising for using old parameters to incrementally learn new knowledge.

+Presentation: The paper writing is good and the idea of paper is easy to follow. Both the figures and tables are easy to understand.

**Weaknesses:**

-In Tables 4 and 5, there are no experimental results for Image-Video, which should be added. Furthermore, I think that the red subscript for the Average results in Table 5 may be unnecessary.

-Tables 4 and 5 show that the final method reduces the $T_2$ performance compared to the method “w/o In-Adapter”. However, the explanation in lines 287-289 is ambiguous, which requires further analysis.

-In Table 3, the metrics of XLLM[22] are missing. Please explain the reasons.

-The authors only provided the comparison on Params and data size, it is better to provide more sufficient analysis to show the efficiency of this continual learning on modalities.

**Questions:**

This method is based on the X-InstructBLIP. If the LLaVA-based method is employed, utilizing a projection layer for modality encoders and LLMs, will this method still be effective? Besides, did the authors verify the generalizability of this method?

**Limitations:**

Yes.

---

> ### Author Rebuttal · Authors · 2024-08-06
>
> Thank you for all your comments. We respond to each of the weaknesses and questions you raised to address your concerns and make the necessary revisions to improve the quality of the paper. Our point-by-point summary and responses to your comments are as follows. (Notes: the mentioned Tables 1-5 are in the original paper, and Table R#PRBt 1-4 are in the attached PDF)
>
> - **Weakness 1: No results for Image-Video and suggestion about removing subscript.**
>
>   - The reason for not presenting the Image-Video results in Tables 4 and 5 is that the results of all ablation methods are identical, **as shown in Table R\#PRBt-1**. Specifically, when extending to the first modality (video), we only add the uni-adapters $\mathcal{A}^1$ without the In-Adapter $\mathcal{F}_{i}^m$ and MoE-based gating modules $\mathcal{G}^m$. The cross-adapters $\hat{\mathcal{A}}^m$ with gating and In-Adapter modules are only employed when extending to the second modality and other modalities. We will provide a more detailed explanation in the revised version to clarify this issue.
>
>   - Furthermore, we have removed the red subscript of Table 5 from the original text.
>
> - **Weakness 2: The performance of the method "w/o In-Adapter" in Tables 4 and 5.**
>
>   $\quad$ In response to this comment, we provide replies and analyses based on Table 4 and Table 5, respectively:
>
>   - **For the $T_{2(in)}$ in Table 4**, the reported score is an average of the AudioCaps Val, AudioCaps Test and AudioCaps QA in Table 5. According to these results of the original paper, we find that there is a small calculation typo in the $T_{2(in)}$, which should be 64.85 $\rightarrow$ 52.47. The revised table **is shown in Table R\#PRBt-2(a)** and the results are corrected in the revised paper. The results indicate that our final model performs best in Table 4.
>
>   - **For the results of AudioCaps Val in Table 5**, we visualize some cases where the final model's scores are much lower than those of the "w/o In-Adapter" model. **In Table R\#PRBt-2(b)**, we show the CIDEr score of these cases obtained by the final model and the "w/o In-Adapter" model, as well as a comparison of their caption results. It can be seen that although the final model has a lower CIDEr score, the answer quality is comparable to the "w/o In-Adapter" model. The final model can describe more objects ```pigeons``` in "Case id: wIvYjuR3nrg" and object ```engine``` in "Case id: yZmhM1HcsyE". In addition, excluding this dataset, the final model's performance on the other 14 datasets and the average result is higher than the "w/o In-Adapter" model.
>
>   $\quad$ Therefore, we believe that these situations do not affect our experimental conclusions. We will add the above related analysis in the revised version to further improve the quality of the paper.
>
>
>
> - **Weakness 3: The metrics of XLLM[22] are missing.**
>
>   $\quad$ There are **two main reasons** why the metrics of X-LLM are missing here:
>
>   - The X-LLM paper does not provide any relevant test results.
>
>   - We attempt to use the released code to test inference results on various datasets, but the model weights are not available.
>
>   $\quad$ Therefore, we mainly want to highlight the trainable parameters and the requirement for joint-modal datasets of X-LLM to showcase the strengths of our method, as illustrated in Table 3. We are trying to rerun the code to supplement the missing results, and we will add the results in the revised version if it can be completed. It may take some time, as the computational resources and time required could exceed 11 days on 8 GPUs. The resource consumption details from the X-LLM paper are listed **in Table R\#PRBt-3**.
>
>
> - **Weakness 4: More analysis about the efficiency of the method.**
>
>   - We further analyze the **Times Cost and GPU Memory** during the training stage between the comparison methods and ours. The relevant results are **shown in Table R\#PRBt-4**. The experiments demonstrate that our method is not only more flexible in extending dataset modalities but also more efficient in terms of training time and memory usage.
>
>
>   - Different methods involve different numbers of modalities and training settings. To ensure fairness, we unify the settings in our evaluation process by using the common modality dataset MSRVTT, the same hyperparameters and training settings: only training on the instruction tuning stage, setting all Batchsize to 4, and keeping the LLMs of BLIP-based X-LLM and ChatBridge frozen.
>
> - **Question: Effectiveness of the proposed modules for LLaVA-based methods and verification of generalizability.**
>
>   $\quad$ Thanks for your inspiring question.
>
>   - For LLaVA-based methods, the proposed modules is effective in terms of anti-forgetting and flexibly expanding modalities for LLaVA-based methods, but it suffers from a significant decline in transfer learning performance. The main reason is that under our setting, the only learnable parameters in LLaVA-based methods are from a linear projection, which is too limited to achieve good transfer performance across multiple modalities. The EProj results of Tables 1 and 2 in the main paper also indirectly proves this situation.
>
>   - Therefore, we mainly focus on the Q-Former-based methods under our setting and do not further explore the our modules' generalizability for other methods, whose training strategies and learnable parameters mostly not be suitable for our setting.
>
>   $\quad$ We believe that this is an interesting question to inspire our future work on how to expand existing pre-trained models to new modalities or tasks with fewer trainable parameters, such as a single linear projection. And we will add a related discussion about this future work in the revised paper.
>
> > [22] Feilong Chen, Minglun Han, Haozhi Zhao, Qingyang Zhang, Jing Shi, Shuang Xu, and Bo Xu. X-llm: Bootstrapping advanced large language models by treating multi-modalities as foreign languages. arXiv preprint arXiv:2305.04160, 2023.

---

> > ### Author Response · Authors · 2024-08-14
> > **Looking forward to your post-rebuttal feedback**
> >
> > Dear Reviewer PRBt:
> >
> > We sincerely appreciate your time and efforts in reviewing our paper. We have provided corresponding responses and results, which we believe have covered your follow-up concerns. We hope to further discuss with you whether or not your concerns have been addressed. Please let us know if you still have any unclear parts of our work.
> >
> > Best,
> >
> > Authors

---

### Author Rebuttal · Authors · 2024-08-06

To all reviewers:

We are grateful to all the reviewers for their valuable comments. We hope that our responses have effectively addressed your previous concerns. We have revised our paper according to your comments. **The major changes are summarized as follows:**

- **According to Reviewer#PRBt Comments:**
  - Experimental Results. We add further analysis and results about the missing Image-Video results in Tables 4 and 5, as shown in Table R#PRBt-1 of attached PDF.

  -  Metrics of XLLM. We provide the explanation for the missing metrics of XLLM and add it in Sec 5.2 of the revised paper.

  - Ablation Study. We correct the results of the $T_{2(in)}$ of Table 4, as shown in Table R#PRBt-2(a) of attached PDF. We give more Q&A visualization of some failure cases in our final model to demonstrate the effectiveness of our final model, as shown in Table R#PRBt-2(b) of attached PDF.

  - Efficiency. We conduct more experiments on the Times Cost and GPU Memory, as shown in Table R#PRBt-4 of attached PDF.


- **According to Reviewer#fRFW Comments:**

  -  Performance Gap. We add more analysis about the advantages of our method compared with fully finetuning method.

  - Related Work "VPGTrans". We summarize the similarities and differences between the VPGTrans and our method, and we add related analysis and discussion in the revised version.

  - Some Issues and Implementation Details. We correct these minor issues and provided more details (Table R#fRFW-1 of attached PDF) about our method in the revised version.

We take this as a great opportunity to improve our work and would appreciate any further feedback you can provide.

Sincerely yours,

Authors.

---

### Decision · Program_Chairs · 2024-09-25

**Decision:**

Accept (poster)

**Comment:**

Thank you for the detailed rebuttal. After reviewing the reviewers' comments and the authors' responses, I recognize the contributions of this paper in introducing continual learning into multimodal large language models (MLLMs). The novel PathWeave with Adapter-in-Adapter (AnA) method presents an innovative approach to expanding the capabilities of pre-trained models across multiple modalities while effectively addressing the challenge of knowledge retention. Given the logical presentation, novel contributions, and comprehensive validation, I recommend accepting this paper.